# Staying on the Manifold: Geometry-Aware Noise Injection

Albert Kjøller Jacobsen[*1], Johanna Marie Gegenfurtner[*1], and Georgios Arvanitidis[1]

[1]Section for Cognitive Systems, DTU Compute, Technical University of Denmark
{akjja, johge, gear}@dtu.dk

## Abstract

It has been shown that perturbing the input during training implicitly regularises the gradient of the learnt function, leading to smoother models and enhancing generalisation. However, previous research mostly considered the addition of ambient noise in the input space, without considering the underlying structure of the data. In this work, we propose several strategies of adding geometry-aware input noise that accounts for the lower dimensional manifold the input space inhabits. We start by projecting ambient Gaussian noise onto the tangent space of the manifold. In a second step, the noise sample is mapped on the manifold via the associated geodesic curve. We also consider Brownian motion noise, which moves in random steps along the manifold. We show that geometry-aware noise leads to improved generalisation and robustness to hyperparameter selection on highly curved manifolds, while performing at least as well as training without noise on simpler manifolds. Our proposed framework extends to data manifolds approximated by generative models and we observe similar trends on the MNIST digits dataset.

**Code:** github.com/albertkjoller/geometric-ml

## 1 Introduction

One of the most intuitive and practical methods to improve the generalisation properties of a learnable model is to consider data augmentation techniques [1]. During training, new data samples are created from given ones, sharing the same features and labels. This approach has been extensively used with e.g. image data, through adjusting the illumination, changing the orientation or cropping.

Classic machine learning research has already established the influence of input noise on generalisation performance [2, 3]. One widely studied technique is adding Gaussian noise to the inputs, which leads to a smoothness penalty on the learnt function [4, 5], however, these works do not take into account the structure of the input data. A fundamental observation in machine learning is the manifold hypothesis: it states that high-dimensional data

---

*Equal contribution. Listed in arbitrary order.

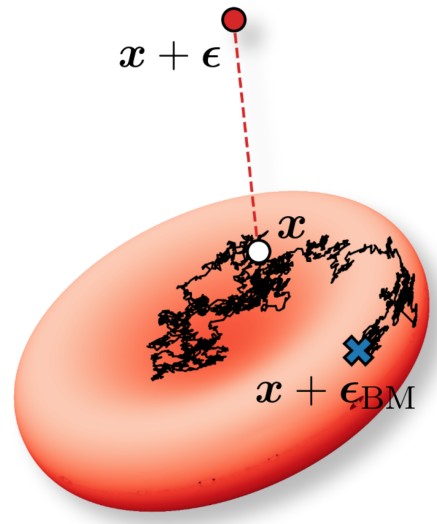

**Figure 1.** Noise injection is a data augmentation technique that can improve generalisation. For a data point ($\bigcirc$) lying on a lower-dimensional manifold, sampling noise in the ambient space ($\bullet$) almost surely deviates from the input manifold whereas a sample from a geometry-aware noise process ($\bm{\times}$) *stays on the manifold* and respects the data geometry. Illustration of the biconcave disc that resembles a red blood cell.

tends to concentrate around a lower-dimensional manifold in the ambient space [6, 7]. In the context of noise-based learning, this has the implication that, with high probability, Gaussian noise will be almost perpendicular to the manifold [8]. Hence, Gaussian input noise gives unlikely or non-informative augmented data samples.

Additionally, many real-world problems require learning functions on a known manifold rather than the unconstrained Euclidean space. Weather and climate observations naturally live on the surface of the sphere, which approximates the shape of the Earth. In cell biology we might consider red blood cells, which can be approximated by a biconcave disc [9]. Or in brain imaging, quantities like cortical thickness and grey matter intensity are measured on the cortical surface [10]: although the cortex can be mapped onto the sphere, it is actually highly wrinkly. In such settings, applying perturbations or learning representations that ignore the intrinsic manifold structure can lead to deceptive results as Euclidean distances in the embedding space fail to capture the

Proceedings of the 7th Northern Lights Deep Learning Conference (NLDL), PMLR 307, 2026.

true distances between points: two points which might be close with respect to the Euclidean metric can be far apart when travelling along the manifold surface. This highlights the necessity of geometry-aware strategies that respect the manifold structure when perturbing data as an augmentation technique.

In this paper, we propose geometry-aware noise injection strategies as a data augmentation technique and show their benefits compared to ambient space noise injection. We consider three such strategies and demonstrate their effect on manifolds embedded in $\mathbb{R}^3$, namely the Swiss roll and families of spheroids and tori. We additionally apply our strategies in the setting of a learnt data manifold, specifically the MNIST digits dataset. Our contributions include:

1. defining geometry-aware input noise for various parameterised, deformed and learned manifolds,

2. establishing the implicit regulariser of adding manifold-restricted input noise,

3. empirical demonstration that geometry-aware noise can improve generalisation and robustness over manifold-agnostic noise.

## 2  Preliminaries

We consider a dataset of $N$ points $\{\boldsymbol{x}_n, \boldsymbol{y}_n\}_{n=1}^N$, where the inputs $\boldsymbol{x}_n \in \mathcal{X} \subseteq \mathbb{R}^D$ are assumed to lie on an embedded $d$-dimensional manifold $\mathcal{M}$ with $d < D$, and the outputs $\boldsymbol{y}_n \in \mathcal{Y}$ may be either continuous or discrete. Our goal is to learn a function $f_{\boldsymbol{\theta}} : \mathcal{X} \to \mathcal{Y}$, typically parameterised by a deep neural network with parameters $\boldsymbol{\theta} \in \mathbb{R}^K$. The model is trained by minimizing the empirical loss

$$\mathcal{L}(\boldsymbol{x}, \boldsymbol{\theta}) = \sum_{n=1}^N \ell(f_{\boldsymbol{\theta}}(\boldsymbol{x}_n), \boldsymbol{y}_n), \qquad (1)$$

where $\ell : \mathcal{Y} \times \mathcal{Y} \to \mathbb{R}^+ \cup \{0\}$ is a loss function, often chosen as the mean squared error (MSE) in regression settings. For simplicity of notation, we write $\boldsymbol{x} = \{\boldsymbol{x}_n\}_{n=1}^N$ and $\mathcal{L}(\boldsymbol{x}) = \mathcal{L}(\boldsymbol{x}, \boldsymbol{\theta})$.

### 2.1  Gaussian Input Noise

Several previous works consider Gaussian input noise [2, 4, 11, 12]. In this section, we summarise the previous analysis and show that adding Gaussian noise to the input during training is equivalent in expectation to Tikhonov regularisation [13].

Consider an input data point $\boldsymbol{x}_n \in \mathcal{X}$, which we perturb with noise following a Gaussian distribution $\boldsymbol{\epsilon} \sim \mathcal{N}(0, \sigma^2 \mathbb{I}_D)$ for $\sigma > 0$. Then the second-order Taylor expansion of the loss function $\mathcal{L}(\boldsymbol{x})$ is:

$$\mathcal{L}(\boldsymbol{x} + \boldsymbol{\epsilon}) \approx \mathcal{L}(\boldsymbol{x}) + \boldsymbol{\epsilon}^\top \nabla_x \mathcal{L}(\boldsymbol{x}) + \frac{1}{2} \boldsymbol{\epsilon}^\top \mathbf{H}_{\mathcal{L}} \boldsymbol{\epsilon}. \quad (2)$$

Taking the expectation of the Gaussian noise yields

$$\mathbb{E}_{\boldsymbol{\epsilon}}[\mathcal{L}(\boldsymbol{x} + \boldsymbol{\epsilon})] = \mathcal{L}(\boldsymbol{x}) + \frac{\sigma^2}{2} \Delta_x \mathcal{L}(\boldsymbol{x}), \qquad (3)$$

where $\Delta_x$ is the Laplace operator (trace of the Hessian) with respect to $x$. When choosing $\ell$ to be the MSE, and using the chain rule, this expands to:

$$\Delta_x \mathcal{L}(\boldsymbol{x}) = \frac{1}{N} \cdot \sum_{n=1}^N \|\nabla_x f_{\boldsymbol{\theta}}(\boldsymbol{x}_n)\|^2 \qquad (4)$$
$$+ \frac{1}{2N} \sum_{n=1}^N (f_{\boldsymbol{\theta}}(\boldsymbol{x}_n) - \boldsymbol{y}_n) \Delta_x f_{\boldsymbol{\theta}}(\boldsymbol{x}_n).$$

When the function interpolates the training data points, that is, $f_{\boldsymbol{\theta}}(\boldsymbol{x}_n) \approx \boldsymbol{y}_n$, the second summand in Equation 4 vanishes[1]. Thus, after plugging this back into Equation 3, we see that adding input noise is equivalent (in expectation) to optimising a regularised loss on the form $\mathcal{L}(\boldsymbol{x}) + R(\boldsymbol{x}, \boldsymbol{\theta})$, with $R$ being the Tikhonov regulariser

$$R(\boldsymbol{x}, \boldsymbol{\theta}) = \frac{\sigma^2}{2N} \sum_{n=1}^N \|\nabla_x f_\theta(\boldsymbol{x}_n)\|^2. \qquad (5)$$

Thus, a small gradient is incentivised at each training point, which implies that the optimisation process will converge to parameters $\boldsymbol{\theta}^*$ for which the function $f_{\boldsymbol{\theta}^*}$ is *flat* in the neighbourhood of the given data.

### 2.2  Riemannian Geometry

**Local charts.**  Plainly speaking, a manifold can be seen as a $d$-dimensional generalisation of a surface. It locally resembles the Euclidean space $\mathbb{R}^d$, meaning that for every point $\boldsymbol{x} \in \mathcal{M}$, we can find an open neighbourhood around $\boldsymbol{x}$ which can be smoothly mapped to an open set of $\mathbb{R}^n$. For completeness, we include a more rigorous mathematical definition.

**Definition 2.1** *A manifold $\mathcal{M}$ is a Hausdorff space such that for every $\boldsymbol{x} \in \mathcal{M}$ there exists a homeomorphism $X : U \to V$ from a neighbourhood $U \ni \boldsymbol{x}$ to an open set $V \subseteq \mathbb{R}^d$. We require these charts to be compatible on the intersection of their domains, i.e.*

$$X_1 \circ X_2^{-1}|_{X_2(U_1 \bigcap U_2)} : X_2(U_1 \cap U_2) \subseteq \mathbb{R}^d \to \mathbb{R}^d$$

*is a smooth map.*

**The tangent space.**  In $\mathbb{R}^3$, the tangent plane of a manifold is easy to picture: each point of the surface is approximated with a plane in which the tangent

---

[1]We assume that $\Delta_x f_\theta(\boldsymbol{x}_n)$ is bounded for all $\boldsymbol{x}_n$ and $\theta$, as $\mathcal{C}^2$-smoothness is satisfied globally for several activation functions, e.g. Softplus and Tanh. Since the set of training points is finite, we conclude that $\Delta_x f_\theta(\boldsymbol{x}_n)$ is bounded. For ReLU architectures, which are not $\mathcal{C}^2$-smooth, the set of cusps has measure zero, and $\nabla_x^2 f_\theta(\boldsymbol{x})$ vanishes almost everywhere.

vectors live. In higher dimensions, we say that the tangent space $T_{\boldsymbol{x}}\mathcal{M}$ of $\mathcal{M}$ at a point $\boldsymbol{x}$ consists of the velocities of all curves on $\mathcal{M}$ passing through $\boldsymbol{x}$, that is, if $\gamma$ is a smooth curve on $\mathcal{M}$ parameterised by time $t$ with $\gamma(0) = \boldsymbol{x}$, then $\boldsymbol{v} = \dot{\gamma}(0) \in T_{\boldsymbol{x}}\mathcal{M}$. Assume we have a smooth parameterisation $X : \mathbb{R}^d \to \mathbb{R}^D$. Then the Jacobian of the chart,

$$\mathbf{J}_X = \left[ \frac{\partial X}{\partial u_1}, \ldots, \frac{\partial X}{\partial u_d} \right] \tag{6}$$

is a function from $\mathbb{R}^d$ to $\mathbb{R}^{D \times d}$ and the tangent space at each point is spanned by the columns of $\mathbf{J}_X$. At every point $\boldsymbol{x} \in \mathcal{M}$, any vector $\boldsymbol{v} \in \mathbb{R}^D$ can be orthogonally decomposed into a tangential and a normal component as $\boldsymbol{v} = \boldsymbol{v}_\top + \boldsymbol{v}_\perp$. In Figure 2, we show a manifold (the sphere) embedded in $\mathbb{R}^3$, and the tangent space at a point.

**Riemannian metrics.** A Riemannian manifold $(\mathcal{M}, g)$ is a smooth manifold equipped with a Riemannian metric. A metric $g$ of $\mathcal{M}$ equips each point $\boldsymbol{x} \in \mathcal{M}$ with an inner product $g_{\boldsymbol{x}}$ on $T_{\boldsymbol{x}}\mathcal{M}$. This tensor field allows us to measure distances and angles on the manifold. Given a smooth parameterisation $X : \mathbb{R}^d \to \mathbb{R}^D$, the matrix valued function

$$\mathbf{J}_X^\top \cdot \mathbf{J}_X : \mathbb{R}^d \to \mathbb{R}^{d \times d} \tag{7}$$

induces a metric. For $X(\boldsymbol{u}) = \boldsymbol{x} \in \mathcal{M}$ and $\boldsymbol{v}, \boldsymbol{w} \in T_{\boldsymbol{x}}\mathcal{M}$, let $\tilde{\boldsymbol{v}}, \tilde{\boldsymbol{w}} \in T_{\boldsymbol{u}}\mathbb{R}^d$ be such that $\mathbf{J}_X \tilde{\boldsymbol{v}} = \boldsymbol{v}$ and $\mathbf{J}_X \tilde{\boldsymbol{w}} = \boldsymbol{w}$. Then the induced metric is

$$g_{\boldsymbol{x}}(\boldsymbol{v}, \boldsymbol{w}) = \boldsymbol{v}^\top \mathbf{J}_X^\top \mathbf{J}_X \boldsymbol{w}. \tag{8}$$

We will often write $g$ to denote the matrix $\mathbf{J}_X^\top \mathbf{J}_X$.

**Geodesics.** A geodesic is locally the shortest path on a manifold. We can write a curve $\gamma : I \subseteq \mathbb{R} \to \mathcal{M}$ on $\mathcal{M}$ as $\gamma(t) = X \circ \alpha(t)$, where $\alpha : I \to \mathbb{R}^d$ is a curve in the parameter space. Then $\gamma$ is a geodesic if and only if $\alpha$ satisfies the following ordinary differential equation (ODE) for all $k = 1, \ldots, d$:

$$\ddot{\alpha}_k(t) = -\sum_{i,j=1}^n \dot{\alpha}_i(t) \dot{\alpha}_j(t) \cdot \Gamma_{ij}^k(\alpha(t)), \tag{9}$$

where $\Gamma_{ij}^k$ denote the so-called Christoffel symbols. It can be shown that if $\mathcal{M}$ is a Riemannian manifold, then for every $\boldsymbol{x} \in \mathcal{M}$ and every unit vector $\boldsymbol{e} \in T_{\boldsymbol{x}}\mathcal{M}$ there exists a unique geodesic $\gamma_{\boldsymbol{e}}$ such that

$$\gamma_{\boldsymbol{e}}(0) = \boldsymbol{x}, \quad \dot{\gamma}_{\boldsymbol{e}}(0) = \boldsymbol{e}. \tag{10}$$

**The exponential map.** One can imagine the exponential map as a function which wraps aluminium foil (the tangent plane) around some object (the manifold). Though the manifold is curved and the tangent space is flat, we can wrap a small part

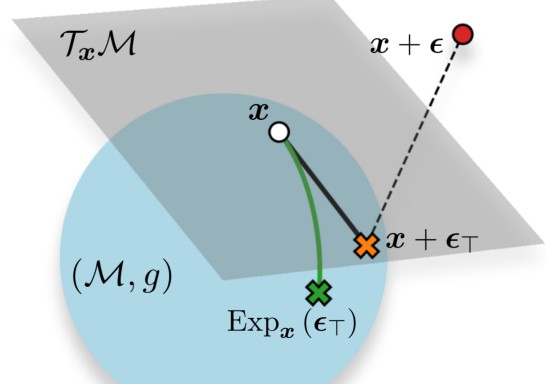

**Figure 2.** Noise injection strategies with increasing level of conceptual complexity, i.e. ambient space noise (●), tangent space noise (✖) and geodesic noise (✖). The Brownian motion strategy is visualised in Figure 3.

of the tangent plane around a neighbourhood of any point without folding the plane.

Using geodesics, for each $\boldsymbol{x} \in \mathcal{M}$ we can define a map from an open ball $B_\delta(0) \subseteq T_{\boldsymbol{x}}\mathcal{M}$ of radius $\delta$ to a neighbourhood $\boldsymbol{x} \in U \subseteq \mathcal{M}$ on the manifold[2], i.e. $\mathrm{Exp}_{\boldsymbol{x}} : B_\delta(0) \subseteq T_{\boldsymbol{x}}\mathcal{M} \to U \subseteq \mathcal{M}$. We will call this map the *exponential map* and define it as:

$$\mathrm{Exp}_{\boldsymbol{x}}(\boldsymbol{v}) = \begin{cases} \gamma_{\frac{\boldsymbol{v}}{\|\boldsymbol{v}\|}}(\|\boldsymbol{v}\|) & \text{if } \boldsymbol{v} \in B_\delta(0) \backslash \{\boldsymbol{0}\}, \\ \boldsymbol{x} & \text{if } \boldsymbol{v} = \boldsymbol{0}. \end{cases} \tag{11}$$

Hence, the exponential map maps a tangent space vector $\boldsymbol{v} \in T_{\boldsymbol{x}}\mathcal{M}$ to the endpoint of a curve on the manifold, $\gamma_{\frac{\boldsymbol{v}}{\|\boldsymbol{v}\|}}(\|\boldsymbol{v}\|)$, and the zero vector to $\boldsymbol{x}$.

# 3 Noise Injection Strategies

We consider three strategies of increasing complexity for geometry-aware input noise: tangential noise, geodesic noise and Brownian motion noise. These noise injection strategies either stay close to the manifold or, better, stay on the manifold.

## 3.1 Projected Tangent Space Noise

One strategy is to project Gaussian noise to the tangent space. This takes a sample in the ambient space, $\boldsymbol{\epsilon} \sim \mathcal{N}\left(\boldsymbol{0}, \sigma^2 \mathbb{I}_D\right)$, and pulls it closer to the manifold. The tangential component $\boldsymbol{\epsilon}_\top$ is found by subtracting the orthogonal part, $\boldsymbol{\epsilon}_\perp$ from $\boldsymbol{\epsilon}$:

$$\boldsymbol{\epsilon}_\top = \boldsymbol{\epsilon} - \boldsymbol{\epsilon}_\perp = \boldsymbol{\epsilon} - \sum_i \langle \boldsymbol{\epsilon}, \boldsymbol{n}_i \rangle \cdot \boldsymbol{n}_i. \tag{12}$$

---

[2]Here, $\delta \in \mathbb{R}^+$ ensures that the exponential map is a well defined diffeomorphism. Loosely speaking, it is the largest radius we can choose while guaranteeing that the geodesics are well defined and do not overlap.

Here, $\{\boldsymbol{n}_i\}$ is a set of unit vectors spanning the normal space of $\mathcal{M}$. For more details we recommend the classic textbook [14]. Equivalently, the tangential noise can be defined as $\boldsymbol{\epsilon}_\top = \mathbf{P}\boldsymbol{\epsilon}$ with projection matrix $\mathbf{P} = \mathbb{I}_D - \sum_i \boldsymbol{n}_i \boldsymbol{n}_i^\top$. This allows for directly sampling tangent noise as $\boldsymbol{\epsilon}_\top \sim \mathcal{N}\left(\mathbf{0}, \sigma^2 \mathbf{P}\right)$.

**Regularisation perspective:** We now analyse how adding tangential noise $\boldsymbol{\epsilon}_\top$ affects the model $f_{\boldsymbol{\theta}}$. We proceed as in Subsection 2.1 and observe that

$$\mathbb{E}[\boldsymbol{\epsilon}_\top^\top \mathbf{H}_{\mathcal{L}} \boldsymbol{\epsilon}_\top] = \frac{1}{N} \sum_{n=1}^N \mathbb{E}[\boldsymbol{\epsilon}_\top^\top \nabla_x f_{\boldsymbol{\theta}}(\boldsymbol{x}_n) \nabla_x f_{\boldsymbol{\theta}}(\boldsymbol{x}_n)^\top \boldsymbol{\epsilon}_\top]$$

$$+ \frac{1}{2N} \sum_{n=1}^N \mathbb{E}[\boldsymbol{\epsilon}_\top^\top \left(f_{\boldsymbol{\theta}}(\boldsymbol{x}_n) - \boldsymbol{y}_n\right) \Delta_x f_{\boldsymbol{\theta}}(\boldsymbol{x}_n) \boldsymbol{\epsilon}_\top] \quad (13)$$

The second summand again vanishes if we assume that the model $f_{\boldsymbol{\theta}}$ interpolates the target values perfectly, that is, $f_{\boldsymbol{\theta}}(\boldsymbol{x}_n) = \boldsymbol{y}_n$ for all $n = 1, \dots, N$. When evaluating the first summand, we use an orthogonal decomposition of the gradient to see that

$$\boldsymbol{\epsilon}_\top^\top \nabla_x f_{\boldsymbol{\theta}}(\boldsymbol{x}_n) = \boldsymbol{\epsilon}_\top^\top \nabla_x f_{\boldsymbol{\theta}}(\boldsymbol{x}_n)_\top + \underbrace{\boldsymbol{\epsilon}_\top^\top \nabla_x f_{\boldsymbol{\theta}}(\boldsymbol{x}_n)_\perp}_{=0}. \quad (14)$$

Combining our results, we obtain the regulariser

$$R(\boldsymbol{x}, \boldsymbol{\theta}) = \frac{\sigma^2}{2N} \sum_{n=1}^N ||\nabla_x f_{\boldsymbol{\theta}}(\boldsymbol{x}_n)_\top||^2. \quad (15)$$

This shows that the addition of tangential noise only regularises the tangential component of $f_{\boldsymbol{\theta}}$.

## 3.2 Geodesic Noise

As explained in Subsection 2.2, at every $\boldsymbol{x} \in \mathcal{M}$, and for every $\boldsymbol{v} \in T_{\boldsymbol{x}}\mathcal{M}$ there exists a geodesic $\gamma : I \to \mathcal{M}$ such that $\gamma(0) = \boldsymbol{x}$, and $\dot{\gamma}(0) = \boldsymbol{v}$. All manifolds in our paper are complete, and hence $I = \mathbb{R}$, and $\gamma$ can be extended to the whole of $\mathbb{R}$. This allows us to generate points $\tilde{\boldsymbol{x}}$ near $\boldsymbol{x}$ by sampling initial velocities and mapping them to the manifold via the exponential map. We proceed as follows: first, sample a velocity $\boldsymbol{\epsilon}_\top$ in the tangent space $T_{\boldsymbol{x}}\mathcal{M}$ as explained in Subsection 3.1, next, evaluate $\gamma$ at $||\boldsymbol{\epsilon}_\top||$ to get the geodesic noise sample,

$$\tilde{\boldsymbol{x}} = \text{Exp}_{\boldsymbol{x}}(\boldsymbol{\epsilon}_\top) = \gamma(||\boldsymbol{\epsilon}_\top||). \quad (16)$$

For a small step size $\sigma$, we expect this to have a similar effect as the tangential noise but may improve robustness for increased step sizes. Details about the implementation can be found in Appendix A.

## 3.3 Intrinsic Brownian Motion

Brownian motion is a stochastic process, which has been used to describe random movement of particles suspended in a fluid. Due to its occurrence in nature,

this provides a realistic way of modelling how data points might move on a manifold. In the parameter space of a Riemannian manifold, Brownian motion is defined by the following stochastic process [15]:

$$du_k(t) = \frac{1}{2} \frac{1}{2\sqrt{\det g}} \sum_{l=1}^d \frac{\partial}{\partial u_l} \left(\sqrt{\det g} \cdot g^{kl}\right) dt$$

$$+ \left(\sqrt{g^{-1}} dB(t)\right)_k \quad (17)$$

where $dB(t)$ is Euclidean Brownian motion and $t$ is the time. The summands are referred to as the drift and noise term, respectively. Since Brownian motion on a manifold is generated by the Laplace-Beltrami operator [16], which is intrinsic, it is independent of the chart [17]. We visualise the strategy in Figure 3.

## 3.4 Example: the Swiss Roll

We will now do the computations for one example manifold, namely the Swiss roll. This manifold is parameterised by a chart $X : \mathbb{R}^2 \to \mathbb{R}^3$ as follows:

$$X(u_1, u_2) = (au_1 \sin u_1, au_1 \cos u_1, u_2).$$

Here, $a \in \mathbb{R}^+$ is a coefficient which determines how tightly the manifold is rolled. The metric $g$ is then

$$g = \text{diag}\left(a^2 \left(1 + u_1^2\right), 1\right).$$

**Tangent space noise.** The unit normal vector at each point $X(u_1, u_2)$ is given by

$$\boldsymbol{n} = \frac{1}{\sqrt{1 + u_1^2}} \cdot \begin{bmatrix} \cos u_1 - u_1 \sin u_1 \\ -\sin u_1 - u_1 \cos u_1 \\ 0 \end{bmatrix}.$$

Following Subsection 3.1, we generate tangential noise from the normal vector and a Gaussian sample.

**Geodesic noise.** A curve on the manifold $\gamma : I \to \mathcal{M}$ can be obtained by taking a curve $\alpha : I \to \mathbb{R}^2$ in the parameter space $\mathbb{R}^2$ and mapping it on the manifold via $X$. For the Swiss roll, the Geodesic Equation, i.e. Equation 9, is equivalent to

$$\ddot{\alpha}_1(t) = -\frac{\alpha_1(t)\dot{\alpha}_1(t)^2}{1 + \alpha_1(t)^2}, \quad \ddot{\alpha}_2(t) = \alpha_2(0) + t\dot{\alpha}_2(0).$$

**Brownian motion.** For the metric $g$, we have

$$\det(g) = a^2(1+u_1^2) \quad \text{and} \quad g^{-1} = \text{diag}\left(\frac{1}{a^2(1+u_1^2)}, 1\right).$$

Plugging these quantities into Equation 17, we get:

$$\begin{bmatrix} du_1 \\ du_2 \end{bmatrix} = -\frac{dt}{2} \begin{bmatrix} \frac{u_1}{(1+u_1^2)^2} \\ 0 \end{bmatrix} + \sqrt{dt} \begin{bmatrix} \frac{1}{\sqrt{a^2(1+u_1^2)}} \\ 1 \end{bmatrix} \odot \tilde{\boldsymbol{\epsilon}}.$$

We remark that $dB(t) = \sqrt{dt} \cdot \tilde{\boldsymbol{\epsilon}}$ where $\tilde{\boldsymbol{\epsilon}} \sim \mathcal{N}\left(\mathbf{0}, \mathbb{I}_d\right)$ is a noise sample in the parameter space.

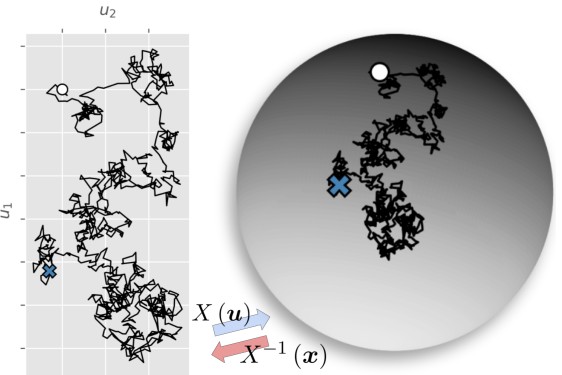

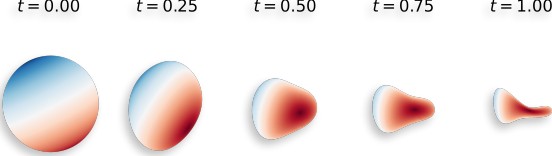

**Figure 4.** The deformation process of the sphere in $\mathbb{R}^3$ for increasing time steps of using a flow field $v_t$.

**Figure 3.** Brownian motion from an initial point ($\bigcirc$) in the parameter space (*left*), and mapped to the manifold (*right*) via the chart $X$. The endpoint of the Brownian motion on the manifold ($\times$) acts as the noisy observation.

with a neural network. Though we in practice only consider a fixed parameterisation of such a network, our framework works for any map $v_t$. This opens new pathways to neural network settings where a learnt flow approximates the data manifold from which we can then compute intrinsic geometric quantities, which we leave for future work.

**Implementation details.** The Jacobian of the vector field, $v_t$, rarely has a closed form, however we can compute it efficiently using *automatic differentiation* (AD) with e.g. JAX or PyTorch. In practice, this allows us to evaluate derivatives of deformed manifolds with respect to the local coordinates of points on the manifold, without manually deriving the expressions. This algorithmic framework allows us to apply the technique to any manifold as long as some parameterisation is available and we have a differentiable ODE solver. In practice, we solve the flow equation numerically using an Euler scheme and compute Jacobians and induced metrics with AD. We remark that higher-order ODE solvers can be used for improved accuracy, yet the Euler scheme was chosen due to challenges with current toolboxes, specifically incompatibility issues between libraries.

# 4 Deformation of a Manifold

We briefly elaborate on an approach to deform parameterised manifolds, which we use in Section 5. We consider a vector field $v$ for defining a time-dependent diffeomorphism, $\phi : \mathcal{M} \times [0, T] \to \mathbb{R}^D$ that maps points from a parameterised manifold to a deformed version of the manifold, $\widetilde{\mathcal{M}}$. This is also known as a *flow*. The vector field $v$ induces the flow through an ordinary differential equation:

$$\phi_0\left(\mathbf{0}\right) = \boldsymbol{x}, \quad \frac{d}{dt}\phi_t\left(\boldsymbol{x}\right) = v_t\left(\phi_t\left(\boldsymbol{x}\right)\right), \quad (18)$$

where $\boldsymbol{x} \in \mathcal{M}$ is a point on the parameterised manifold. We can then express points on the deformed manifold through the local coordinates of the parameterised manifold as $\tilde{\boldsymbol{x}} = \phi_T\left(X\left(\boldsymbol{u}\right)\right) \in \widetilde{\mathcal{M}}$, which is obtained by integrating the ODE up to time $T$. We provide an illustration of such a deformation process for the sphere in Figure 4. The Jacobian of $\phi_t$ with respect to $u$ at $\boldsymbol{u} = X^{-1}(\boldsymbol{x})$ is given by

$$\mathbf{J}_{\boldsymbol{u}}\left(t\right) := \frac{\partial \phi_t\left(X\left(\boldsymbol{u}\right)\right)}{\partial u} = \frac{\partial \phi_t\left(\boldsymbol{x}\right)}{\partial x}\frac{\partial X\left(\boldsymbol{u}\right)}{\partial u}.$$

It can be computed by solving another ODE:

$$\mathbf{J}_{\boldsymbol{u}}\left(0\right) = \frac{\partial X\left(\boldsymbol{u}\right)}{\partial u}, \quad \frac{d}{dt}\mathbf{J}_{\boldsymbol{u}}\left(t\right) = \mathbf{J}_v\left(t\right)\mathbf{J}_{\boldsymbol{u}}\left(t\right), \quad (19)$$

where $\mathbf{J}_v\left(t\right) := \frac{\partial v_t\left(\phi_t\left(\boldsymbol{x}\right)\right)}{\partial \phi_t}$ is the Jacobian of the velocity field function. Thus, the metric $\tilde{g}$ of $\widetilde{\mathcal{M}}$ is

$$\tilde{g} = \mathbf{J}_{\boldsymbol{u}}\left(T\right)^{\top}\mathbf{J}_{\boldsymbol{u}}\left(T\right). \quad (20)$$

This allows sampling vectors on the tangent space $T_{\tilde{\boldsymbol{x}}}\widetilde{M}$ at $\tilde{\boldsymbol{x}}$ and generating geodesics or Brownian motion on the deformed manifold $\widetilde{M}$ by pulling the metric back to the parameter space. This framework allows for highly expressive and flexible deformations of any parameterised manifold while ensuring invertibility. Previous research [18, 19] parameterise $v_{t,\boldsymbol{\theta}}$

# 5 Experimental validation

## 5.1 Parameterised Manifolds in $\mathbb{R}^3$

We first test our hypothesis on a range of parameterised manifolds in $\mathbb{R}^3$. We generate $N = 200$ training points on each manifold and train an overparameterised 3-layer neural network with 64 nodes per layer to learn a specific function for each manifold. We train for 500 epochs using a learning rate of $10^{-3}$ with a MSE objective. For the `DeformedSphere` we only use $N = 40$ and a learning rate of 0.005 for computational speed-up. For each training step, we add either ambient space noise, tangential noise, geodesic noise or Brownian motion noise and compare to a baseline network trained without adding input noise. We treat the noise covariance $\sigma^2$ as a hyperparameter, and, in the Brownian motion setting, interpret it as the total time of the process, i.e. $T = \sigma^2$. We provide the average error per strategy relative to the baseline's MSE in Table 1 with uncertainties given

**Table 1.** Mean squared error relative to the baseline (B) model trained without adding noise. We report results for the optimal hyperparameter $\sigma^2$ for each strategy and manifold. We compare with ambient noise (A), tangent noise (T), geodesic noise (G) and Brownian motion noise (BM). We highlight the best strategy per manifold in **bold**. Adding noise does not improve performance for some manifolds, but results are included for completeness. We include illustrations of the manifolds and functions on manifolds that we consider. The deformation approach described in Section 4 is used to construct the `DeformedSphere` from a parameterised unit sphere in $\mathbb{R}^3$.

| | Sphere | SqueezedSphere | DeformedSphere | Bead | OnionRing | SwissRoll |
|---|---|---|---|---|---|---|
| B | $1.00 \pm 0.16$ | $1.00 \pm 0.15$ | $1.00 \pm 0.26$ | $1.00 \pm 0.09$ | $\mathbf{1.00 \pm 0.19}$ | $1.00 \pm 0.18$ |
| A | $\mathbf{0.91 \pm 0.10}$ | $1.01 \pm 0.15$ | $1.08 \pm 0.26$ | $0.99 \pm 0.08$ | $1.24 \pm 0.24$ | $1.00 \pm 0.19$ |
| T | $0.98 \pm 0.14$ | $\mathbf{0.94 \pm 0.17}$ | $1.10 \pm 0.23$ | $1.00 \pm 0.09$ | $1.13 \pm 0.24$ | $0.62 \pm 0.07$ |
| G | $1.00 \pm 0.16$ | $1.01 \pm 0.16$ | $1.00 \pm 0.25$ | $0.99 \pm 0.08$ | $1.10 \pm 0.21$ | $0.47 \pm 0.06$ |
| BM | $1.00 \pm 0.16$ | $0.96 \pm 0.18$ | $\mathbf{0.92 \pm 0.23}$ | $\mathbf{0.98 \pm 0.09}$ | $1.13 \pm 0.18$ | $\mathbf{0.46 \pm 0.06}$ |

Manifold: 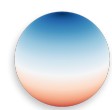 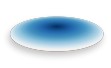 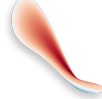 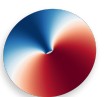 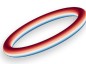 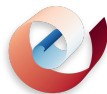

by the standard error of the mean computed from 5 independent runs. We provide computations for the geodesic equation and Brownian motion along with the target function for each manifold in Appendix B.

Our results show that geometry-aware noise injection provides advantages to ambient space noise on complex manifolds. In particular, geodesic and Brownian motion noise yield lower errors on "wigglier" geometries, such as the `SwissRoll`, and they also exhibit greater robustness to the noise intensity hyperparameter (Figure 5). This indicates that geometric approaches can both improve generalisation and reduce sensitivity to hyperparameter choices. At the same time, performance rarely significantly deteriorates when using any noise strategy, compared to the baseline trained without noise (Table 1). For some manifolds, simple ambient Gaussian noise can suffice, particularly for those of which only a small part is problematic, such as the `Bead` (the fat torus). Here, Gaussian noise only leads to misleading samples near the genus. Since the surface area of the genus is proportionally small, the overall error remains low. The `SwissRoll`, on the other hand, is sensitive to Gaussian noise everywhere, and our strategies work better. For completeness, we report results across all manifolds, even when geometry-aware strategies do not provide measurable gains.

**Which is the better strategy?** Though both geodesic noise and Brownian motion noise perform equally well under certain conditions, Brownian motion noise is computed more efficiently than geodesic noise, which requires solving the exponential map with high precision. Due to the stochastic nature of Brownian motion, it is less affected by the resolution of the time discretisation which allows for speeding up the sampling process. For these reasons, we restrict further analyses to only consider our geometry-aware Brownian motion strategy.

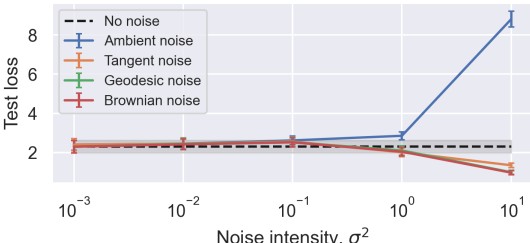

**Figure 5.** Test loss on the `SwissRoll` as a function of noise intensity $\sigma^2$ for different noise injection strategies. The geometry-aware noise strategies that stay on the manifold, i.e. geodesic noise and Brownian motion noise, show greater robustness to the noise intensity compared to ambient or tangential noise. Our strategies perform at least as well as training without noise (dashed line).

## 5.2 MNIST

We now turn our attention to a higher-dimensional example using image data, where the manifold must be approximated. One common approach to approximate such a manifold is using autoencoders [20]. An autoencoder uses an encoder-decoder structure to reconstruct input data samples with minimum reconstruction error. As such, an autoencoder has an inherent latent space $\mathcal{Z} \subseteq \mathbb{R}^d$ in which we can represent the data samples using the encoder, i.e. $z = f_e(x)$. The reconstruction is obtained by decoding the latent representation to a point on the approximate data manifold $\widetilde{\mathcal{X}} \subseteq \mathbb{R}^D$, i.e. $\tilde{x} = f_d(z)$. We can therefore think of the latent space $\mathcal{Z}$ as the parameter space of the approximate data fold $\widetilde{\mathcal{X}}$ with the decoder serving as the chart (similar to Figure 3). Typically, $d \ll D$ which makes it favourable for doing manipulations of the data and defining the pullback metric of the approximated data manifold in the latent space allows us to apply our geometry-aware noise injection strategies on approximated data manifolds.

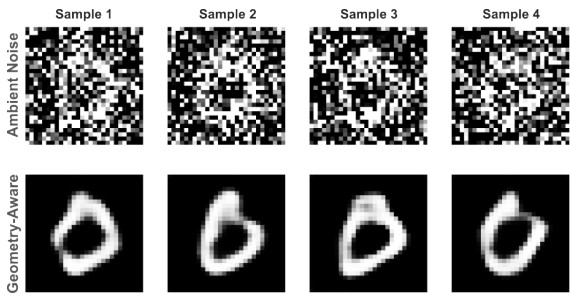

**Figure 6.** Four different augmentations of a specific image of a '0' under ambient Gaussian noise (*top*) and geometry-aware Brownian motion noise (*bottom*) for a relatively large noise intensity of $\sigma = 1$. The geometry-aware samples resemble digits and thus *stay on the manifold* which is not the case for the ambient noise samples.

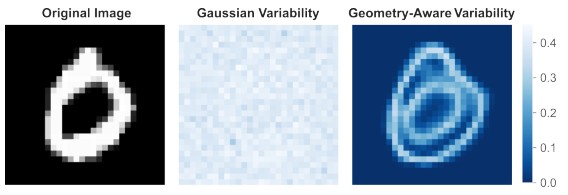

**Figure 7.** An example image of a '0' (*left*) along with the pixel-wise standard deviation over 100 augmentations of it, using ambient Gaussian noise (*middle*) and intrinsic Brownian motion (*right*). Where ambient noise is somewhat uninformative, the geometry-aware noise targets natural variations on the edges of the digit.

We test our hypothesis using intrinsic Brownian motion on the approximated image manifold, specifically on the MNIST digits dataset. First, we train an autoencoder on the full training set. Next, we train a 1-layer MLP classifier with 1024 hidden units with various levels of Brownian motion noise added to the data during training. We compare to adding Gaussian noise in the ambient image space. The MNIST dataset consists of 60,000 samples and covers the image manifold of digits well, for which reason we test our strategy in settings of subsampling the training data to 1%, 10% or 50% of the dataset. We do so to examine highly overparameterised settings where data augmentation is expected to improve the model fit. See Appendix C for experimental details.

In Figure 6, we show different augmentations with ambient noise and intrinsic Brownian motion noise for an example of a '0'. While our geometry-aware approach generates digit-looking images, the underlying signal is hard to recognise in the case of Gaussian noise. In Figure 7 we show the associated pixel-wise variations across 100 augmented samples for each method. While the ambient noise variation is somewhat uniform, the geometry-aware samples give natural variations along the edges of the digit.

In the most overparameterised setting using 1% of the data, our geometry-aware noise injection

**Table 2.** Test accuracy on MNIST when trained on the original images (O), reconstructed images (R), original images with ambient noise (A) and reconstructed images with geometry-aware noise (BM). The header refers to the subsampling rate of the training set and uncertainties are the standard errors over 10 independent runs. We highlight the best performance per subsampling rate in bold. Green cells indicate whether training with noise is significantly better than training on the original or reconstructed images for the respective noising strategies.

|     | 1%                  | 10%                 | 50%                 |
| --- | ------------------- | ------------------- | ------------------- |
| O   | $0.883 \pm 0.008$   | $0.956 \pm 0.002$   | $\mathbf{0.981 \pm 0.001}$ |
| A   | $0.883 \pm 0.008$   | $\mathbf{0.965 \pm 0.001}$ | $\mathbf{0.981 \pm 0.001}$ |
| R   | $0.877 \pm 0.005$   | $0.943 \pm 0.002$   | $0.967 \pm 0.002$   |
| BM  | $\mathbf{0.896 \pm 0.008}$ | $0.959 \pm 0.002$ | $0.971 \pm 0.001$   |

strategy shows improved performance over learning without noise and learning with ambient Gaussian noise (see Table 2). In this setting, we additionally see that increasing the noise intensity of the ambient noise deteriorates the classifier's performance, while the trend is opposite for the geometry-aware noise strategy (Figure 9). It is worth noting the performance gap of approximately 0.6% when trained on the original images compared to the reconstructed images, yet we highlight that the geometry-aware noise eventually surpass this gap.

When the classifier is trained on larger amounts of the training set, the performance gap between training on the original and reconstructed images grows, resulting in the geometry-aware strategy not improving over training without noise on the original images. Yet, we note that our strategy performs consistently better than the classifier trained directly on the reconstructed images. We therefore expect the strategy to work well if lowering the autoencoder's approximation error, i.e. learning a better approximation of the data manifold. We remark that learning a perfect approximation of the data manifold is not the aim of this paper.

One potential limitation of the geometry-aware strategy is that the augmented samples might resemble other digits than the label associated with the original sample. This is due to the fact that the strategy does not have information about the digit labels from the decoder itself. If the intrinsic Brownian motion crosses the label boundary, it can negatively impact the classifier performance due to label noise. For intuition, see the example of transitioning from a '4' to a '9' in Figure 8. We considered solving this potential issue by also pulling back information from the classifier activations to the latent space, however initial experiments revealed no significant performance gain. This could be due to the fact that the augmented samples already lie along the label border, giving a stronger and more robust classifier.

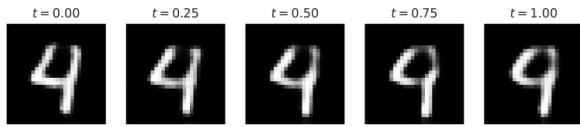

**Figure 8.** Time slices of the geometry-aware Brownian motion starting from an image of a '4' at time $t = 0$. As the decoder is unaware of the digit labels, the Brownian motion can cross the label boundary, resulting in the augmentation at time $t = 1$ resembling a '9'.

# 6   Related Work

A recent work [3] surveys classical perspectives and modern advances for how noise injection influences learning. Instead of assuming that the input points live on a manifold, we can also enforce that the parameters of the model belong to a manifold. A previous work [5] analyses the impact of adding Gaussian noise to weights of a parametric model. Other works [21, 22] study orthogonal regularisers on the weight matrices, promoting the columns to be orthonormal. These constraints restrict the parameter space to the Stiefel or Grassmann manifolds, which improves numerical stability. This line of work highlights that geometry can be injected not only through noise in the input space but also by shaping the structure of the model's parameters. Other works inject noise to the gradient during training with gradient-based optimisers for improved generalisation [23, 24].

In the context of Riemannian representation learning, adding noise according to the structure of the manifold stabilises results in the recent paper [25]. This approach replaces the traditional encoder–decoder setup with a Riemannian generative decoder. It directly optimises manifold-valued latent variables via a Riemannian optimiser, thereby avoiding the difficulties of approximating densities on complex manifolds. By enforcing the manifold structure during training, the learnt latent representations remain aligned with the intrinsic geometry of the data, leading to more interpretable models and stable training dynamics.

In a recent work [26], the tangent plane of a data manifold is approximated through singular value decomposition and used for sampling points in alignment with the data structure, similar to our tangent space noise. For the methodology of the geodesic noise, a related idea has been explored in the context of Riemannian Laplace approximations for Bayesian inference in deep neural networks [27, 28].

# 7   Conclusion

We have established several geometry-aware noise injection strategies and demonstrated their need

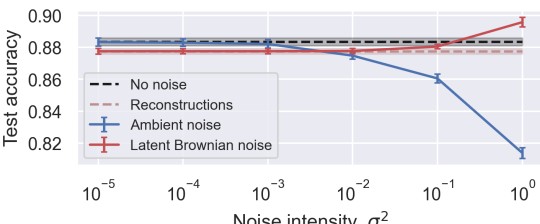

**Figure 9.** Test accuracy of an overparameterised 1-layer MLP trained on the 1% subsampled MNIST dataset using ambient noise and our proposed geometry-aware Brownian motion strategy. For ambient noise, the model performance deteriorates, while geometry-aware Brownian motion improves generalisation.

through theoretical and experimental contributions. Further, we have shown their qualities and shortcomings. In particular, we find that while ambient Gaussian noise is simple and may improve performance on nearly Euclidean manifolds, it falls short on more curved or "wiggly" manifolds, where geodesic and Brownian motion noise provide clear advantages. These geometry-aware strategies not only improve generalisation, but are also more robust to the noise intensity with the latter reducing the burden of hyperparameter tuning. We proposed a framework for deforming parameterised manifolds to arbitrary manifolds, which extends the use of our strategies beyond standard benchmark geometries. However, we remark that this added flexibility currently comes with increased computational cost. Lastly, we showed an how to apply our techniques to higher-dimensional manifolds approximated by an autoencoder.

**Limitations and future work.** Though our results in the high-dimensional setting of image data did not give strictly better performance, we attributed the performance gap to the quality of the manifold approximation, thus future work involves learning a better approximator of the manifold using e.g. flow matching as established in Section 4. We expect a large difference between the dimensions of the ambient space and the data manifold to lead to more dramatic results, as Gaussian noise samples will with high probability be normal to the manifold. Thus, studying the relation between the ambient space dimensionality, data manifold dimensionality and the classifier performance is of interest.

# Acknowledgments

This work was supported by Danish Data Science Academy, which is funded by the Novo Nordisk Foundation (NNF21SA0069429) and VILLUM FONDEN (40516), and by the DFF Sapere Aude Starting Grant "GADL". The Otto Mønsted Fond provided generous support for the authors' travel. We also want to thank the reviewers for the helpful feedback.

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

# A  Implementation Details

## A.1  Geodesic Noise

To simplify our computations, instead of sampling a vector $\boldsymbol{\epsilon}_\top \sim \mathcal{N}\left(0, \sigma^2 \mathbf{P}\right)$ in the tangent space, we can also sample a vector $\tilde{\boldsymbol{\epsilon}}$ in the parameter space $\mathbb{R}^d$ from an adjusted distribution. In the following assume that $\boldsymbol{u} \in \mathbb{R}^d$, $X(\boldsymbol{u}) = \boldsymbol{x} \in \mathcal{M}$, where $X$ is a smooth parameterisation of a regular manifold $\mathcal{M}$. As previously described, the Jacobian transforms vectors in the parameter space to the tangent space, i.e. for a vector $\boldsymbol{\epsilon} \in T_{\boldsymbol{u}} \mathbb{R}^d$, we have that

$$\boldsymbol{\epsilon} = \mathbf{J}_X \tilde{\boldsymbol{\epsilon}} \in T_{\boldsymbol{x}} \mathcal{M}.$$

For the inverse relation, we obtain

$$\tilde{\boldsymbol{\epsilon}} = g^{-1} \mathbf{J}_X^\top \boldsymbol{\epsilon}.$$

Consequently, if

$$\boldsymbol{\epsilon} \sim \mathcal{N}(0, \sigma^2 \mathbb{I}_D),$$

then for its tangential component it holds that

$$\boldsymbol{\epsilon}_\top \sim \mathcal{N}(0, \sigma^2 \mathbf{P}),$$

and for the pullback it holds that

$$\tilde{\boldsymbol{\epsilon}}_\top \sim \mathcal{N}\left(\mathbf{0}, \sigma^2 g^{-1} \mathbf{J}_X^\top \mathbf{P} \mathbf{J}_X g^{-1}\right), \qquad (21)$$

which follows from affine transformation properties of the multivariate Gaussian distribution.
This allows us to find the curve $\alpha : \mathbb{R} \to \mathbb{R}^d$ such that

$$\alpha(0) = X^{-1}(\boldsymbol{x}), \quad \dot{\alpha}(0) = \tilde{\boldsymbol{\epsilon}}_\top.$$

Our new sample point is then

$$\tilde{\boldsymbol{x}} = X\left(\alpha(\|\tilde{\boldsymbol{\epsilon}}_\top\|)\right).$$

This strategy is equivalent to the one described in Subsection 3.2. For simplicity, we ignore the injectivity radius of the domain of the exponential map – this is not a problem since we do not require injectivity for our purposes, and the manifolds we consider are complete.

## A.2  Functions on the Manifolds

For the Sphere, SqueezedSphere and DeformedSphere, we select the target function as

$$y = v,$$

that is, the second local coordinate.

For the Bead we select the target function as

$$y = \sin v$$

which is a periodic function of the second local coordinate.

For the OnionRing we select the target function as:

$$y = 100 \cdot c \cdot \cos u = 100 \cdot z,$$

which is the scaled height of the manifold.

For the SwissRoll we select the target function as:

$$y = u,$$

namely the first local coordinate.

# B  Manifold Computations

## B.1  Biconcave disc

The biconcave disc yields an approximation of human erythrocytes, as shown in [9]. Letting $r = \sqrt{u^2 + v^2}$, and let $a, b, c, d$ be parameters, then the height function for the upper half is given by

$$z(r) = d\sqrt{1 - \frac{4r^2}{d^2}} \cdot \left(a + \frac{br^2}{d^2} + \frac{cr^4}{d^4}\right).$$

Here, $d$ describes the diameter, $a$ the height at the centre, $b$ the height of the highest point, and $c$ the flatness in the centre. A parameterisation of the upper half of this surface of rotation is given by

$$X(r, \theta) = (r\cos\theta, r\sin\theta, z(r)).$$

### B.1.1  Tangential noise on the biconcave disc

The tangent space is then spanned by

$$X_r = \left[\cos\theta, \sin\theta, \frac{\partial z}{\partial r}\right],$$
$$X_\theta = [-r\sin\theta, r\cos\theta, 0].$$

A standard computation shows that

$$\frac{\partial z}{\partial r} = \frac{-8r}{d\sqrt{1 - \frac{4r^2}{d^2}}} \cdot \left(a + \frac{br^2}{d^2} + \frac{cr^4}{d^4}\right)$$
$$+ \sqrt{1 - \frac{4r^2}{d^2}} \cdot \left(\frac{2br}{d} + \frac{4cr^3}{d^3}\right),$$

and clearly

$$\frac{\partial r}{\partial u} = \frac{2u}{r}, \quad \frac{\partial r}{\partial v} = \frac{2v}{r}.$$

The unit normal vector is now given by

$$\boldsymbol{n} \;=\; \frac{\left[\frac{\partial z}{\partial r} r \cos\theta, -\frac{\partial z}{\partial r} r \sin\theta, r\right]}{r\left(\frac{\partial z}{\partial r}^2 + 1\right)}.$$

### B.1.2 Geodesics on the biconcave disc

We obtain

$$g(r,\theta) = \begin{pmatrix} 1 + \frac{\partial z}{\partial r}^2 & 0 \\ 0 & r^2 \end{pmatrix}.$$

A computation shows that

$$\ddot{r}(t) \;=\; -\frac{\frac{\partial z}{\partial r}\frac{\partial^2 z}{\partial r^2}}{1 + \frac{\partial z}{\partial r}^2} \cdot \dot{r}(t)^2 + \frac{r(t)}{1 + \frac{\partial z}{\partial r}^2} \cdot \dot{\theta}(t)^2,$$

$$\ddot{\theta}(t) \;=\; -\frac{2\dot{r}(t)}{r(t)} \cdot \dot{\theta}(t).$$

The second derivative of $z$ is given by the following:

$$\frac{\partial^2 z}{\partial r^2} \;=\; \frac{-4}{d}\left(1 - \frac{4r^2}{d^2}\right)^{-\frac{3}{2}}\left(a + \frac{br^2}{d^2} + \frac{cr^4}{d^4}\right)$$

$$-\frac{16r}{d}\cdot\left(1 - \frac{4r^2}{d^2}\right)^{-\frac{1}{2}}\cdot\left(\frac{br}{d^2} + \frac{2cr^3}{d^4}\right)$$

$$+2\sqrt{1 - \frac{4r^2}{d^2}}\cdot\left(\frac{b}{d^2} + \frac{6cr^2}{d^4}\right).$$

### B.1.3 Brownian motion on the biconcave disc

For the Brownian motion, we have that

$$dr(t) \;=\; \frac{1}{2}\cdot\left(\frac{1 + \frac{\partial z}{\partial r}^2 - \frac{\partial z}{\partial r}\frac{\partial^2 z}{\partial r^2}}{(1 + \frac{\partial z}{\partial r}^2)^2}\right) dt$$

$$+\frac{1}{\sqrt{1 + \frac{\partial z}{\partial r}^2}} dB(t)_1,$$

$$d\theta(t) \;=\; \frac{1}{2}\cdot\left(\frac{r\frac{\partial z}{\partial r}\frac{\partial^2 z}{\partial r^2} - 1 - \frac{\partial z}{\partial r}^2}{r^3(1 + \frac{\partial z}{\partial r}^2)}\right) dt + \frac{1}{r} dB(t)_2,$$

for all $r > 0$.

## B.2 Spheroids

We consider manifolds which are squeezed spheres. For $a, c \in \mathbb{R}^+$, consider the parameterisation $X : \mathbb{R}^2 \to \mathbb{R}^3$ given by

$$X(u,v) \;=\; (a\sin u\sin v, a\sin u\cos v, c\cos u).$$

If $a = c$, then this gives the usual sphere. If $a > c$, then the manifold is a sphere squished along the $z$-axis. The tangent plane is spanned by

$$X_u \;=\; [a\cos u\sin v, a\cos u\cos v, -c\sin u],$$
$$X_v \;=\; [a\sin u\cos v, -a\sin u\sin v, 0].$$

We then obtain the metric

$$\begin{pmatrix} a^2\cos^2 u + c^2\sin^2 u & 0 \\ 0 & a^2\sin^2 u \end{pmatrix}.$$

### B.2.1 Tangential noise on the spheroid

To obtain tangential noise, we note that the unit normal is given by

$$\boldsymbol{n} = \frac{[c\sin u\sin v, c\sin u\cos v, a\cos u]}{\sqrt{c^2\sin^2 u + a^2\cos^2 u}}.$$

### B.2.2 Geodesics on the spheroid

A curve $\gamma = X \circ \alpha$ is a geodesic on the spheroid if and only if $\alpha : I \to \mathbb{R}^2$ satisfies

$$\ddot{\alpha}_1(t) \;=\; \frac{(a^2 - c^2)\sin\alpha_1(t)\cos\alpha_1(t)}{a^2\cos^2\alpha_1(t) + c^2\sin^2\alpha_1(t)} \cdot \dot{\alpha}_1(t)^2$$

$$+ \frac{a^2\sin\alpha_1(t)\cos\alpha_1(t)}{a^2\cos^2\alpha_1(t) + c^2\sin^2\alpha_1(t)} \cdot \dot{\alpha}_2(t)^2,$$

$$\ddot{\alpha}_2(t) \;=\; -2\cdot\frac{\cos\alpha_1(t)}{\sin\alpha_1(t)}\dot{\alpha}_1(t)\dot{\alpha}_2(t).$$

### B.2.3 Brownian motion on the spheroid

A computation yields the following result for the Brownian motion.

$$du_k(t) \;=\; \left[\begin{array}{c} \frac{a^2\cos u}{2\sin u(a^2\cos^2 u + c^2\sin^2 u)^2} \\ 0 \end{array}\right]_k dt$$

$$+ \left[\begin{pmatrix} \frac{1}{\sqrt{a^2\cos^2 u + c^2\sin^2 u}} & 0 \\ 0 & \frac{1}{a\sin u} \end{pmatrix} dB(t)\right]_k.$$

## B.3 Tori

We also investigate different tori, some more like onion rings, others more like beads. For coefficients $a, c \in \mathbb{R}^+$, they can be parameterised by $X : \mathbb{R}^2 \to \mathbb{R}^3$,

$$X(u,v) = ((a + c\sin u)\sin v, (a + c\sin u)\cos v, c\cos u).$$

Here, $c$ describes the thickness of the handle and $a$ the size of the torus. To avoid self-intersection, $c$ is bounded by $a$. Further, if $c << a$, we have an onion ring, and if $c \uparrow a$ we have a rounded torus with a very thin hole.

### B.3.1 Tangential noise on tori

The tangent plane is spanned by

$$X_u \;=\; [c\cos u\sin v, c\cos u\cos v, -c\sin u],$$
$$X_v \;=\; [(a + c\sin u)\cos v, -(a + c\sin u)\sin v, 0].$$

This yields the metric

$$g = \begin{pmatrix} c^2 & 0 \\ 0 & (a + c\sin u)^2 \end{pmatrix}.$$

The unit normal is given by

$$\boldsymbol{n} = \frac{[\sin u\sin v, \sin u\cos v, \cos u\sin^2 v]}{\sqrt{\sin^2 u + \cos^2 u\sin^2 u}}.$$

### B.3.2 Geodesic noise on tori

A curve $\gamma = X \circ \alpha$ on the torus is a geodesic if and only if $\alpha$ satisfies

$$\ddot{\alpha}_1(t) \;=\; \frac{(a + c\sin\alpha_1(t))\cos\alpha_1(t)}{c} \cdot \dot{\alpha}_2(t)^2,$$

$$\ddot{\alpha}_2(t) \;=\; 2\frac{c\cos\alpha_1(t)}{a + c\sin\alpha_1(t)} \cdot \dot{\alpha}_1(t)\dot{\alpha}_2(t).$$

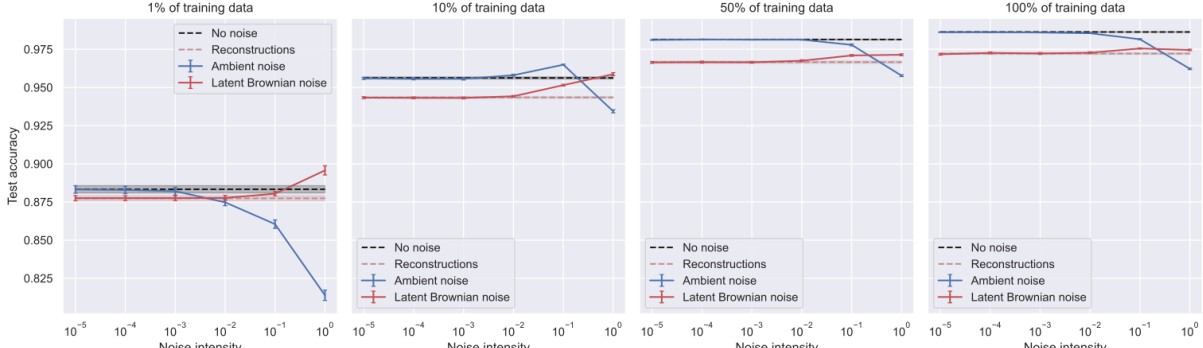

**Figure B.1.** Test accuracy of an overparameterised 1-layer MLP trained on different subsampled versions of the MNIST dataset using ambient noise and our proposed geometry-aware Brownian motion strategy.

### B.3.3 Brownian motion on tori

We obtain the Brownian motion terms

$$
\begin{aligned}
du_k(t) \;=\; & \left[ \begin{array}{c} \frac{\cos u}{2c(a+c\sin u)} \\ 0 \end{array} \right]_k dt \\
& + \left[ \left( \begin{array}{cc} \frac{1}{c} & 0 \\ 0 & \frac{1}{a+c\sin u} \end{array} \right) dB(t) \right]_k
\end{aligned}
$$

## C MNIST Experiment

We first trained an autoencoder on the full training dataset. Both the encoder and decoder were defined as convolutional neural networks with softplus activation functions and a $d = 16$ dimensional latent space. For stability, we choose the output function of the decoder to be the hyperbolic tangent. Since the hyperbolic tangent maps the real line to $(-1, 1)$, and we considered MNIST images normalized to the pixel range of $[0, 1]$, we convert the decoder output to lie in the same pixel interval. We train the autoencoder with the MSE loss objective using a batch size of 64, a learning rate of 0.01 and weight decay of $3 \cdot 10^{-5}$ for 100 epochs using Adam and a cosine learning rate schedule acting every epoch.

Next, we train a classifier to distinguish the MNIST digits using different versions of the images: 1) the original images, 2) the reconstructed images using the autoencoder, 3) the images with ambient Gaussian noise and 4) the images based on geometry-aware Brownian motion in the latent space. For Gaussian noise in the ambient space, we clip the pixel-values to the $[0, 1]$ range. We do so using either 1%, 10%, 50% and 100% of the training dataset. We define the classifier as a 1-layer MLP with 1024 hidden units using ReLU as the activation function. We use a learning rate of 0.001 and train the classifier until convergence for 100 epochs with Adam and a cosine learning rate scheduler acting every epoch. We use the negative log-likelihood as the training objective. We repeat the experiment for noise intensities $\sigma$ log-spaced between $10^{-4}$ and 1.

All training was repeated for 10 different random initialisations of both the autoencoder and the classifier. We compute the mean accuracy on the test set for each noise intensity along with the standard error of the mean

**Table C.1.** Test set accuracy for the best performing classifiers trained on the full training dataset.

|      | 100%              |
| ---- | ----------------- |
| Or   | $0.986 \pm 0.001$ |
| A    | $\mathbf{0.986 \pm 0.001}$ |
| Re   | $0.972 \pm 0.002$ |
| BM   | $0.976 \pm 0.001$ |

over these 10 runs. We show the results for all settings in Figure B.1. In Table 2 and C.1 we show the best test accuracy (i.e. for the best noise intensity) when training the classifier on all subsampled versions.

## D Manifold Deformation

Recall the definition of the flow field from Equation 18:

$$
\frac{d}{dt}\phi_t\left(X\left(\boldsymbol{u}\right)\right) = v_t\left(\phi_t\left(X\left(\boldsymbol{u}\right)\right)\right).
$$

We take the derivative with respect to the local coordinates $\boldsymbol{u}$ and get

$$
\frac{\partial}{\partial u}\left(\frac{d}{dt}\phi_t\left(X\left(\boldsymbol{u}\right)\right)\right) = \frac{\partial}{\partial u} v_t\left(\phi_t\left(X\left(\boldsymbol{u}\right)\right)\right),
$$

which, assuming continuous second partial derivatives, is equivalent to

$$
\frac{d}{dt}\frac{\partial}{\partial u}\phi_t\left(X\left(\boldsymbol{u}\right)\right) = \frac{\partial}{\partial u} v_t\left(\phi_t\left(X\left(u\right)\right)\right).
$$

By using the chain rule on the right hand side, we get

$$
\frac{d}{dt}\frac{\partial}{\partial u}\phi_t\left(X\left(\boldsymbol{u}\right)\right) = \left.\frac{\partial v_t\left(\phi_t\left(\boldsymbol{x}\right)\right)}{\partial \phi_t}\right|_{\boldsymbol{x}=X(\boldsymbol{u})} \frac{\partial}{\partial u}\phi_t\left(X\left(\boldsymbol{u}\right)\right).
$$

We get the Jacobian ODE by setting

$$
\begin{aligned}
\mathbf{J}_{\boldsymbol{u}}(t) \;&:=\; \frac{\partial \phi_t\left(X\left(\boldsymbol{u}\right)\right)}{\partial u}, \\
\mathbf{J}_v(t) \;&:=\; \left.\frac{\partial v_t\left(\phi_t\left(\boldsymbol{x}\right)\right)}{\partial \phi_t}\right|_{\boldsymbol{x}=X(\boldsymbol{u})}.
\end{aligned}
$$

