# OpenReview forum: "Staying on the Manifold: Geometry-Aware Noise Injection"
_NLDL.org/2026/Conference — NLDL 2026 Poster_

### Official Review · Reviewer_srDo · 2025-10-03
**report**

**Rating:** 4
**Confidence:** 3

**Summary:**

It is known that adding noise to the input during training might lead to implicit regularization and enhancing the generalization performance. However, many real-world applications require learning functions on a given manifold rather than a generic Euclidean space. The author(s) proposed geometry-aware noise injection for various parameterized manifolds and their deformations, and studied the implicit regularization.  Numerical results were provided to demonstrate better generalization performance and robustness over manifold-agnostic noise.

**Strengths:**

(1) The problem is interesting and well motivated.

(2) The three structured injected noise schemes introduced in the paper are reasonable, and the analysis is appropriate.

**Weaknesses:**

(1) The example discussed in Section 3.4. seems to have low dimensions. It would be great if you can have some high-dimensional and large-scale examples.

(2) There should be more discussions comparing the three different noise injection methods you proposed in Sections 3.1., 3.2. and 3.3. It is not clear at the moment for applications, which one performs better than the others, and when one should use one rather than the other two.

(3) In Sections 3.1., 3.2. and 3.3., you proposed three different noise injection schemes. I suppose in practice, the structured noise is harder to implement than unstructured noise. It would be great if some discussions can be added about numerical efficiency of implementing the injected noises in Sections 3.1., 3.2. and 3.3., especially intrinsic Brownian motion in Section 3.3., which is not clear to me that it is easy to implement in practice when the dimension is high and the number of data points is huge.

(4) Some notations should be better presented. For example, you mentioned that $\Delta$ is the Laplace operator after equation (2), but in equation (2) what you used was actually $\Delta_{x}$. So you should explain that $\Delta_{x}$ is the Laplace operator w.r.t. $x$ for example. Also, for example, in the second paragraph of Section 3.1., you never defined $\epsilon_{\perp}$.

**Justification:**

Overall, it is a well-written paper. Injecting structured noise is motivated by applications, and is an important problem to study. The methodologies proposed in the paper are reasonable and appropriate. What lacks is a more detailed discussion between the different structured noise injections proposed in the paper, and it will also be better if some higher-dimensional and larger-scale examples
are provided to further illustrate the methodologies proposed in the paper.

---

> ### Author Rebuttal · Authors · 2025-10-22
>
> We thank the reviewer for the effort and time taken to evaluate our work, for the positive consideration of our work and the constructive comments to the content. We address all your concerns below.
>
> > "It would be great if you can have some high-dimensional and large-scale examples."
> - Indeed, including such examples would add value to the paper. We picked simple examples to rigorously establish the theory and to show how we translate the methods from theory into practice. As we mention in section 4, we propose a deformation framework for applying the concepts to unknown data manifolds. While this would indeed also work for higher-dimensional data manifolds, a similar approach could be to approximate the data manifold using LVMs (as also proposed by reviewer pAVX). Doing so, we can consider the latent space as the local coordinate space in $\mathbb{R}^d$ and use the decoder as the chart $X: \mathbb{R}^d \rightarrow \mathbb{R}^D$ that parametrises the data manifold. If accepted, we plan to include such an example on image data.
>
> > "There should be more discussions comparing the three different noise injection methods... It is not clear at the moment for applications, which one performs better than the others, and when one should use one rather than the other two."
> - That is an excellent suggestion. While our results suggest that Brownian motion noise works well in most of the simplest cases, we will go more into details about performance for the specific manifolds and why/why not the methods work well, in the camera-ready version. For applications, one can choose either of the three but there are computational benefits of using Brownian motion noise over geodesic noise (elaborated slightly below).
>
> > "It would be great if some discussions can be added about numerical efficiency of implementing the injected noises ..."
> - Thanks for pointing this out. To give a brief summary, intrinsic Brownian motion noise is computationally faster than geodesic noise because the geodesic technique requires solving the exponential map (which is computationally more heavy). We will add a discussion of this, if accepted.
>
> > "... is not clear to me that it [red. Browian motion] is easy to implement in practice when the dimension is high and the number of data points is huge."
> - In practice, our implementation works for any manifold (high- or low-dimensional) as long as a parametrisation of the manifold is available. If so, computing the metric using the Jacobians is surprisingly easy through automatic differentiation frameworks (e.g. JAX, PyTorch) and the same goes for the step-wise Brownian motion update in Equation 7. In fact, these quantities can also be computed with finite difference method at the cost of increased computation time. The intended experiment on image data with LVMs should reflect that the approach is rather straight-forward to implement, also in high dimensions.
>
> > "Some notations should be better presented..."
> - This will be fixed of course. Thanks for pointing out the specific instances where our notation is unclear or imprecise.

---

### Official Review · Reviewer_pAVX · 2025-10-06

**Rating:** 4
**Confidence:** 2
**Final Rating:** 4
**Final Confidence:** 2

**Summary:**

This paper introduces a set of geometry-aware noise injection strategies for training machine learning models on data manifolds, addressing the limitation that conventional noise injection in the ambient space often disregards the data’s intrinsic geometric structure. The authors develop and analyze three distinct approaches - tangent-space noise, geodesic noise, and Brownian motion noise - providing theoretical insights into their regularization effects and conducting extensive empirical evaluations across a variety of parameterized and deformed manifolds. The findings demonstrate that geometry-aware noise not only enhances model generalization and robustness but also improves hyperparameter stability, particularly in settings involving highly curved manifolds.

**Strengths:**

Solid Theoretical Analysis: The manuscript provides a comprehensive mathematical derivation of the effects of geometry-aware noise (tangent, geodesic, Brownian motion) on loss function regularization. Sections 2 and 3 carefully establish how classical input noise leads to Tikhonov regularization in ambient space and extend this reasoning to intrinsic methods; the theoretical link between tangent noise and regularization is clear and well-motivated.

Clarity and Exposition: The writing is generally clear and organized, with necessary background in Riemannian geometry covered for non-expert readers (Section 2.2). Definitions, assumptions, and mathematical expressions are transparent and easy to follow. The methodology section outlines each proposed noise process step-by-step, and implementation details are sufficiently thorough to support reproducibility (Section 4, paragraph "Implementation details").

Figures and Visualizations: Figures are well-integrated and clarifying.

Significance: The work addresses an important and often overlooked issue: that conventional noise injection in high-dimensional learning may fail to yield meaningful augmentations because of the manifold hypothesis, and offers concrete, analytical, and empirical remedies. The methods generalize to a wide array of parametrized manifolds (Section 4) and are directly relevant to practitioners training models on data with intrinsic geometric structure.

**Weaknesses:**

Could the authors provide further empirical evaluation on real datasets (e.g., images or scientific data) where the data manifold is estimated via autoencoders or other generative methods, not just explicitly parametrized toy models? If such results are promising, this would greatly strengthen the significance claims.

**Final Justification:**

The authors have addressed my concerns. I am keeping my rating unchanged.

**Justification:**

N/A

---

> ### Author Rebuttal · Authors · 2025-10-22
>
> We thank the reviewer for the effort and time taken to evaluate our work and for the positive consideration of our work. We address your concerns below.
>
> > "Could the authors provide further empirical evaluation on real datasets (e.g., images or scientific data) where the data manifold is estimated via autoencoders or other generative methods, not just explicitly parametrized toy models? If such results are promising, this would greatly strengthen the significance claims."
> - Indeed, this would greatly strengthen our claims and we plan to extend our work fully to real data in a follow up paper where our current work focuses on a rigorous description and justification of our approaches. It is a good idea to consider estimating the data manifold of real datasets. The idea of using LVMs is very promising as we can directly leverage the latent space as the local coordinate space $\mathbb{R}^d$ and the decoder as the chart $X: \mathbb{R}^d \rightarrow \mathbb{R}^D$.
> - If accepted, we plan to include such examples on image data. With this experiment, we would essentially have a two-step modelling pipeline of 1) approximating the data manifold with the LVM and 2) training a classifier on data in image space where we add geometry-aware noise using Brownian motion in the latent space. The only requirement different to the low-dimensional manifolds considered so far, is that the LVM approximates well the true image data manifold. The rest of the modeling is identical to what we did for the lower-dimensional synthetic problems.

---

### Official Review · Reviewer_JDRu · 2025-10-08
**Theoretical framing is sound, the constructions are clear, and the synthetic results align with the intuition that geodesic/Brownian perturbations help on highly curved manifolds.**

**Rating:** 4
**Confidence:** 3

**Summary:**

This work introduces geometry-aware input noise injection strategies for learning on data that lie on a lower-dimensional manifold embedded in an ambient Euclidean space. Beyond classic ambient Gaussian perturbations, the authors develop three manifold-respecting schemes: (i) projecting Gaussian noise onto the tangent space, (ii) mapping tangent noise to the manifold via the exponential map, and (iii) simulating intrinsic Brownian motion on the manifold. They provide derivations on parametrised manifolds and a deformation framework via diffeomorphic flows.

**Strengths:**

The work is conceptually clean and well-motivated. The theoretical treatment connects ambient Gaussian noise to Tikhonov regularisation and extends this view to tangential regularisation, clarifying \emph{what} is being penalised in expectation. Methodologically, the paper strikes a good balance between rigor and practicality. Also, the deformation framework broadens applicability beyond textbook manifolds, hinting at a path to learned manifolds and more realistic settings.

**Weaknesses:**

The evaluation is limited to small synthetic regression tasks on low-dimensional manifolds embedded in $\mathbb{R}^3$, leaving open how the methods behave at realistic scales.
The paper convincingly demonstrates potential on didactic setups but would benefit from validation on real datasets where the manifold is only approximately known or must be learned.

**Justification:**

This is a well-written and timely contribution that formalises and tests geometry-aware input noise for manifold-structured data.

---

> ### Author Rebuttal · Authors · 2025-10-22
>
> We thank the reviewer for the effort and time taken to evaluate our work and for the positive consideration. We address your concerns below.
>
> > "The evaluation is limited to small synthetic regression tasks on low-dimensional manifolds embedded in
> $\mathbb{R}^3$... The paper ... would benefit from validation on real datasets where the manifold is only approximately known or must be learned."
> - We agree that our initial experiments were conducted on low-dimensional datasets with known manifolds to enable precise evaluation and clear interpretation of the method’s behaviour under controlled conditions. This setting allows us to isolate and analyse specific properties without confounding factors.
> - We intend to explore the extension to real data fully in a follow-up work, while our current work focuses on a rigorous description of the noise injection techniques as well as justifying how these techniques can be applied in cases where the manifold must be learned, e.g. through our deformation framework. As proposed by reviewer pAVX, a straightforward idea is to approximate the data manifold using a LVM.
> - If accepted, we plan to add such examples on image data. Essentially, this allows for using the latent space of the LVM as the local coordinate space $\mathbb{R}^d$, hence parametrising the image manifold in $\mathbb{R^D}$ using the decoder as the chart $X: \mathbb{R}^d \rightarrow \mathbb{R}^D$. This procedure is identical to what we did for the smaller manifolds, apart from approximating the data manifold with a neural network and possibly increased computational complexity due to increasing the dimensionality.

---

### Official Review · Reviewer_xjoU · 2025-10-09
**This paper does not convey any meaningful contribution at all.**

**Rating:** 1
**Confidence:** 3

**Summary:**

This paper proposes a data augmentation by adding geometry-aware noise to the training data. Specifically, the paper describes three different strategies – tangent space noise, geodesic noise, and Brownian motion noise – to add a noise vector that reflects the underlying manifold structure. The paper also describes the manifold deformation from a parameterized manifold to another deformed version. Using experiments on artificial datasets, the paper tries to demonstrate improvements in the mean squared error when learning functions on manifolds.

**Strengths:**

I don’t find any strength in this paper.

**Weaknesses:**

1. This paper’s practical contribution is not clear. The introduction states that “data augmentation is extensively used with image data,” but this paper does not directly tackle the image domain.
2. This paper also mentions some problems associated with learning functions on known manifolds; however, the experiment does not utilize any example problems from those real-world domains.
3. The practical value is very limited to specific cases where the manifold is explicitly known. This is very unlikely in practice, and many required components, such as the dimension of the manifold, the tangent space, and the exponential map, are unknown.
4. The authors may argue that, in a real-world scenario, it is possible to approximate those components with cutting-edge manifold learning techniques. However, the authors are responsible for both providing an implementation (using manifold learning on real-world data) and demonstrating the performance. Before the authors provide empirical evidence that the proposed methods have value in real-world applications, this paper does not make any practical contribution.
5. The authors derive some conclusions regarding regularization with Gaussian noise, specifically Equation 4. First of all, the authors cannot simply discard the second term as long as there is no proper assumption on the Laplace operator, such as boundedness.
6. Additionally, from Equation 4, the authors conclude that the optimization will converge to a function with a flat neighborhood around the training data. However, such a case is too ideal and does not happen in practice. In adversarial machine learning, adding Gaussian noise is not an effective defense against adversarial examples. It is usually possible to find an adversarial example near the training data, which indicates a large gradient value near the given data, after the training converges.
7. All the experiments are performed on low-dimensional input data whose underlying manifold is known. These experiments do not represent any practical scenario, e.g., high-dimensional data whose underlying manifold is completely unknown, so the results will not generalize to high-dimensional data in practice.
8. The functions defined on the manifolds are too simple and cannot represent the complexity of real-world problems.
9. The experimental results are not impressive. Table 1 shows that, on half of the datasets, the proposed method did not improve by more than 0.1. This indicates that the proposed method can only achieve marginal improvements, even on low-dimensional datasets.
10. There are a few minor writing issues, such as grammatical errors and a missing equation reference. However, these are not serious issues compared to the main weaknesses of this paper.

**Justification:**

Overall, I don’t think this paper meets the minimum quality standards for a conference paper. The motivation of the paper is unclear, and I still don’t know what benefit the authors expect from the geometric-aware noise. As mentioned in the weaknesses, the paper lacks practical value, and the authors did not describe how the proposed method can be generalized to real-world applications. If the paper only focuses on the low-dimensional domain with known data manifolds, its impact is too weak to be accepted in an academic conference. Also, the claimed contributions are doubtful. The theoretical result has a flaw in its derivation, and the experiment demonstrates only marginal improvement, even in low-dimensional data experiments.

---

> ### Author Rebuttal · Authors · 2025-10-22
>
> We thank the reviewer for the effort and time taken to evaluate our work. We address all your comments in the following.
>
> > "I don’t find any strength in this paper."
> - The main focus of our work is to extend the widely studied Euclidean Gaussian noise model, commonly used for theoretical analysis of implicit biases in deep networks, to the Riemannian manifold setting. To the best of our knowledge, no prior work provides a similar technical framework; most existing approaches are largely empirical.
>
> > "The introduction states that “data augmentation is extensively used with image data,” but this paper does not directly tackle the image domain."
> - Indeed, the main goal of the paper is not data augmentation in the image domain, but we just used it as a motivating example. In the updated version of the paper we clarify this further, but we will also include experiments in the image domain.
>
> > "the experiment does not utilize any example problems from those real-world domains."
> - We agree that examples with real-world data would add further value to our findings. However, we prioritized rigorously describing and justifying the proposed approach, as well as formulating this theoretical contribution in a pedagogical manner. Indeed, our initial experiments were conducted on low-dimensional datasets with known manifolds to enable precise evaluation and clear interpretation of the method’s behaviour under controlled conditions. This setting allows us to isolate and analyse specific properties without confounding factors.
> - As far as we understand, based on the general review criteria listed on the conference website, the absence of real-world data does not determine the overall significance of a paper.
>
> > "The practical value is very limited to specific cases where the manifold is explicitly known. This is very unlikely in practice, and many required components, such as the dimension of the manifold, the tangent space, and the exponential map, are unknown"
> - Indeed, we consider manifolds in $\mathbb{R}^3$ where the parametrisation is known. Our noise injection techniques are however not limited to such settings, as we also mention in the paper. Our framework extends to any case where we know or can approximate a parametrisation of the manifold - one such example is the deformation framework for approximating an unknown data manifold, that we propose in section 4. Another approach is to first learn an LVM for approximating the data manifold, e.g. on image data, which directly induces a Riemannian metric in the latent space that we can utilize for our noise injection approaches (tangential, geodesic and BM).
>
> > "The authors may argue that, in a real-world scenario, it is possible to approximate those components with cutting-edge manifold learning techniques. However, the authors are responsible for both providing an implementation (using manifold learning on real-world data) and demonstrating the performance."
> - Indeed, providing such an example would add value to our paper, and in the camera-ready version we will include such experiments on image data using LVMs to capture the data manifold geometry. Learning an unknown data manifold is indeed a popular and hard research topic, but the great feature of our noise injection techniques is that they are in general agnostic to the specific manifold learning approach. In addition, our proposed deformation framework that builds upon flow matching shows that we can apply our techniques even with the modern advanced generative methods that approximate the data manifold in real-world problems.
>
> > "The authors derive some conclusions regarding regularization with Gaussian noise, specifically Equation 4. First of all, the authors cannot simply discard the second term as long as there is no proper assumption on the Laplace operator, such as boundedness."
> - Our derivations are in line with numerous previous works [4, 5, 12]: see the classic paper Bishop (1995), or a more recent work by Rifai (2018). These works usually assume that the model interpolates the data perfectly, i.e. $f_\theta(x_n)-y_n = 0$ and hence the second term vanishes. If however $f_\theta(x_n)-y_n \approx 0$, as we wrote, you are completely right. For completeness, we will include this assumption of boundedness of the Laplace operator in the paper.
>
> > "... the authors conclude that the optimization will converge to a function with a flat neighbourhood around the training data. However, such a case is too ideal and does not happen in practice. In adversarial machine learning, adding Gaussian noise is not an effective defense against adversarial examples."
> - The main goal of the paper is not to provide a defence mechanism for adversarial attacks, but rather the extension of Euclidean Gaussian noise to the manifold setting and analyse the associated inductive biases.
> - Indeed, adding Euclidean Gaussian noise *in practice* will not make the function flat for several reasons. For example, in high dimensions, the Euclidean Gaussian noise will be with high-probability orthogonal to the underlying data manifold, which implies that a lot of optimisation time will be spent inefficiently. Instead, geometric noise injection focuses on the relevant (learnt) data manifold.
> - Also, the classic result in Equation 4 describes an implicit regulariser which shows that the model is incentivised to learn a function that is flat in the neighbourhood of the given data, i.e. how input noise would *theoretically* affect the loss. Here we mean that the function is merely regularised, as opposed to being forced to be flat.
>
> > "experiments do not represent any practical scenario, e.g., high-dimensional data whose underlying manifold is completely unknown, so the results will not generalize to high-dimensional data in practice."
> - Extending the methods to high-dimensional data is indeed a valid direction that we plan to further explore in the future. As mentioned, our methods works for high-dimensional data as long as we can parametrise the data manifold, e.g. by approximating it with an LVM or using the proposed deformation framework. We never claim that the results will generalise to high-dimensional data, however under the manifold hypothesis and the curse of dimensionality, we expect that respecting the underlying manifold will be superior to adding ambient Gaussian noise.
>
> > "The functions defined on the manifolds are too simple and cannot represent the complexity of real-world problems."
> - Indeed we could have considered more complex functions on the manifolds. However, these controlled scenarios allow us to analyse what makes our methods work better than adding ambient space noise to the data. Studying synthetic problems is a general approach to gaining intuition on how different methods behave.
>
> > "The experimental results are not impressive... the proposed method can only achieve marginal improvements, even on low-dimensional datasets."
> - Although this is the case for some manifolds, like the sphere, it is not the case for other manifolds like the SwissRoll, where the difference between the Euclidean and geodesic distance measures is large.
>
> > "There are a few minor writing issues, such as grammatical errors and a missing equation reference."
> - We will include references for all equations and proofread our manuscript; however, we would appreciate any pointers regarding the grammatical errors you are referring to.

---

### Meta-Review · Area_Chair_ASKd · 2025-10-31

**Recommendation:** Reject
**Confidence:** 3

**Metareview:**

The views on this paper are very mixed with both positive and negative assessments. There is no consensus among the reviewers, and the author rebuttal has not changed this. I generally concur with the negative assessment that the paper neither presents actual theoretical or empirical contributions, and that the the experiments are performed in too limited cases to argue for real-world impact. Based on all the reviews, having to make a recommendation, I see strongest arguments for the negative assessment leading to a recommendation of rejecting the paper.

---

### Decision · Program_Chairs · 2025-11-05

**Decision:**

Accept (Poster)

**Comment:**

Based on the authors' comments and rebuttal, and given the positive reviews of three authors, we recommend the acceptance of the work as a poster.